# Emulgel with *Origanum vulgare* L. Oil: A New Therapeutic Proposal in Case of Dermal Bacterial Infections

**DOI:** 10.3390/ph18111768

**Published:** 2025-11-20

**Authors:** Mariana Ganea, Diana Constanța Pelea, Florina (Miere) Groza, Octavia Gligor, Laura Grațiela Vicaș, Marcel Zdrîncă, Antonia Maria Lestyan, Marieta Lestyan, Ionuț Daniel Venter, Mădalin Florin Ganea, Laura Maghiar, Timea Claudia Ghitea, Corina Moisa

**Affiliations:** 1Pharmacy Department, Faculty of Medicine and Pharmacy, University of Oradea, 1st December Square 10, 410028 Oradea, Romania; mganea@uoradea.ro (M.G.); timea.ghitea@csud.uoradea.ro (T.C.G.); corinamoisa@hotmail.com (C.M.); 2Preclinic Department, Faculty of Medicine and Pharmacy, University of Oradea, 1st Decembrie Street, 410028 Oradea, Romania; diana_pelea@uoradea.ro (D.C.P.); octavia.gligor@uoradea.ro (O.G.); antonia.lestyan@uoradea.ro (A.M.L.); marietalestyan@yahoo.com (M.L.); 3Surgical Disciplines Department, Faculty of Medicine and Pharmacy, University of Oradea, 1st December Square 10, 410028 Oradea, Romania; marcelzdrinca@yahoo.ro; 4Clinical County Emergency Hospital Oradea, 65 Gh.Doja Street, 410169 Oradea, Romania; laura.maghiar@uoradea.ro; 5Doctoral School of Biomedical Sciences, University of Oradea, 1 Universitătii Street, 410073 Oradea, Romania; venterionutdoctorat@gmail.com; 6Faculty of Medicine and Pharmacy, University of Oradea, 1st December Square 10, 410028 Oradea, Romania; ganea.madalinflorin@student.uoradea.ro; 7Psychoneuroscience and Recovery Department, University of Oradea, 1st Decembrie Street, 410028 Oradea, Romania

**Keywords:** *Origanum vulgare* L. oil, emulgel formulation, antimicrobial capacity, cutaneous reaction, in vivo testing

## Abstract

**Background**: The treatment of bacterial dermatological diseases is currently facing major difficulties, determined by the alarming increase in the resistance of pathogenic bacteria to conventional therapies. In this context, a viable and effective alternative is represented by the use of phytocompounds to obtain the desired therapeutic effect. The essential oil of *Origanum vulgare* L. stands out for its antibacterial, anti-aging, collagen synthesis stimulating and wound healing properties. However, its use is limited by certain disadvantages, such as poor stability and the risk of skin irritation due to accumulation in the dermis. **Method**: The process of formulating the emulgel with oregano oil respected the specific technological steps. The resulting emulgel was subjected to a series of tests, including organoleptic, stability and antimicrobial efficacy determinations. In addition, an in vivo study was conducted to confirm the lack of irritation, involving six groups of patients differentiated by age, sex and skin phenotype. **Results**: The test results revealed that the emulgel formulated with oregano oil is stable, has organoleptic properties and an appropriate pH for topical use. The product demonstrated antibacterial efficacy against *Staphylococcus aureus* and *Pseudomonas aeruginosa*. In addition, short-term in vivo tests (20 min—96 h) confirmed the safety and absence of skin irritation, indicating its potential as an effective alternative treatment. **Conclusions**: In conclusion, the emulgel with origanum oil represents an innovative formulation for topical application. The product is well tolerated by the skin and does not cause irritation, and its antibacterial properties validate it as a promising therapeutic solution.

## 1. Introduction

Bacterial dermatological diseases are an increasing therapeutic challenge due to widespread bacterial resistance to conventional treatments. Consequently, innovative pharmaceutical approaches based on phytocompounds are gaining importance as effective alternatives. Among aromatic plants, those rich in volatile oils—derived from roots, stems, leaves, flowers, or whole plants—offer complex phytochemical profiles with both primary metabolites and bioactive secondary compounds. *Origanum vulgare* L. (oregano), a member of the *Lamiaceae* family, is particularly valued for its essential oil rich in terpenoid compounds such as monoterpenes, sesquiterpenes, flavonoids, phenolic acids, and tannins, which contribute to antimicrobial, antioxidant, and wound-healing activities. However, the high volatility and dermal irritancy of these compounds limit their direct topical use, making optimized delivery systems necessary to ensure stability and safety [1,2,3].

The phytochemical composition of the volatile oil obtained from *Origanum vulgare* L. is well known in the specialized literature, highlighting important compounds such as monoterpenes, sesquiterpenes, flavonoids, phenolic acids, and tannins. These compounds are directly responsible for the positive pharmacological effects in the dermatological sphere when applied to the skin, so that oregano oil can be used in the treatment of acne, in the case of dermal infections, having a strong antimicrobial effect, as an antioxidant, anti-aging, or healing agent. It should be noted, however, that volatile oils have the ability to remain in the dermis for a longer time, thus having the possibility of becoming irritating to the skin [4].

The major disadvantages of volatile oil compounds also include instability to environmental factors such as light, temperature, and humidity. For this reason, its use in dermatological preparations with respect to obtaining the expected effect is conditioned by the type of pharmaceutical formula [5].

To reduce these undesirables, the careful choice of the pharmaceutical formula in which the volatile oil is incorporated is a key factor in the success of the treatment plan, so that its inclusion in a transport vesicle (liposome, microcapsule, micro- or nano-sphere) or in a three-dimensional network is a recommended approach [6].

The inclusion of volatile oil compounds in a therapeutic system such as emulgel is a practical and efficient pharmaceutical approach for dermal application. The inclusion of the emulsion containing the volatile oil in a gelling system increases its stability and bioavailability at the skin level, reducing the effects of its accumulation and irritability. The emulgel presents a controlled, sustained and significantly superior release compared to classic formulas for topical application of the compounds present in the volatile oil [7,8].

The aim of our study is to formulate an emulgel with oregano oil to be tested for antimicrobial activity on bacteria frequently found on the skin in the case of various dermatological conditions.

We consider that the inclusion of oregano volatile oil in a three-dimensional network that is formed in the case of emulgel preparation is a novelty, as this pharmaceutical preparation has not existed until now.

Another novelty of this work is represented by the in vivo testing of the innovative pharmaceutical preparation, from the point of view of irritability, using a complex statistical analysis.

The oregano essential oil used in this study was obtained by steam distillation from flowering aerial parts of *Origanum vulgare* L., a method that preserves volatile monoterpenes such as carvacrol and thymol while minimizing thermal degradation. Steam distillation is the most commonly applied technique for *Origanum vulgare* oil extraction, as it ensures reproducibility of the chemotype and bioactivity profile. This type of extraction, together with its controlled origin (Albania, 2024 harvest), supports the pharmacological consistency of the material used in our emulgel formulation.

## 2. Results

### 2.1. Organoleptic Control of Emulgel with Oregano Oil (OvO)

The OvO emulgel (emulgel with oregano oil) obtained has a homogeneous appearance, white color and characteristic odor. Its consistency is a characteristic of the emulgel formula, being semi-solid with a gelled, viscous appearance.

In addition, the measured pH of the OvO gel is slightly acidic (pH = 6), being considered safe for application to the skin. The immediate stability of the emulgel was tested by centrifugation, with the emulgel remaining homogeneous. The stability index (SI) value for the OvOF emulgel (emulgel without oregano oil) is 85%, and for the OvO emulgel, it is 95%, which highlights the high stability of the prepared formula. The OvO emulgel has a higher stability index value than OvOF, which can be attributed to the inclusion of oregano oil that, through its composition, improves the stability of the “gel” network of the formulated emulgel.

### 2.2. Results of the Antimicrobial Capacity of OvO Emulgel

The bacterial capacity testing of the formulated emulgel (OvOF and OvO) was performed on the Gram-positive bacteria *Staphylococcus aureus* and the Gram-negative bacteria *Pseudomonas aeruginosa*. The results obtained are highlighted in Table 1.

Although our study focused on emulgel formulations, the literature data provides a useful benchmark for comparison. Pure *Origanum vulgare* L. essential oil exhibits MIC values between 0.25 and 0.5 mg/mL against *S. aureus* and *P. aeruginosa* [9]. The OvO emulgel in the present work achieved a comparable antibacterial effect within 1–2 h of exposure while using an oil concentration of only 0.5%, indicating enhanced efficacy and faster onset due to improved dispersion and sustained release within the emulgel matrix. In contrast, direct application of the oil at equivalent concentrations is reported to cause dermal irritation and reduced homogeneity. Therefore, incorporation into the emulgel system appears to optimize both antimicrobial activity and skin compatibility.

### 2.3. Statistical Results

#### 2.3.1. Demographic Characteristically Description

A total of six experimental groups (Lots 1.00 to 6.00), each comprising 34 participants, were analyzed with respect to gender distribution, age, skin phototype (Fitzpatrick classification), and responses to three time-dependent tests (Test A at 20 min, Test B at 24 h, and Test C at 96 h). All continuous variables were tested for normality using the Shapiro–Wilk test. Skin response scores were normally distributed, justifying the use of parametric tests. Tests A, B, and C represent objective measurements of erythema intensity taken at 20 min, 24 h, and 96 h, respectively, using a standardized visual analog scale (VAS) (Table 2).

Lots 1.00 through 4.00 consisted exclusively of female participants, whereas Lots 5.00 and 6.00 included only males. The mean age varied across groups, ranging from 20.59 ± 3.49 years in Lot 1.00 to 61.79 ± 10.25 years in Lot 3.00 among females, and from 24.47 ± 5.80 years in Lot 5.00 to 54.06 ± 13.75 years in Lot 6.00 among males.

The intensity of response measured by Test A (20 min post-application) and Test B (24 h) showed group- and age-dependent variability. In female groups, response magnitude tended to increase with age: Lot 1.00 had a Test A mean of 1.06 ± 2.04, while Lot 3.00 exhibited the highest Test A score of 2.13 ± 3.15. A similar pattern was observed for Test B (0.29 ± 1.19 in Lot 1.00 vs. 1.03 ± 2.05 in Lot 3.00). Male participants in Lot 5.00 demonstrated the lowest response levels (Test A: 0.88 ± 1.93; Test B: 0.15 ± 0.86), while those in Lot 6.00, who were older, showed moderately increased scores (Test A: 1.00 ± 1.94; Test B: 1.03 ± 2.05). Notably, no measurable response was recorded at 96 h in any group (Test C = 0.00 ± 0.00 across all lots), suggesting complete resolution or absence of prolonged effects.

Skin phototype distribution revealed a predominance of Fitzpatrick phototype II across all groups, ranging from 44.1% to 61.8%. Phototype III was more prevalent in older or male groups, reaching up to 38.2% in Lots 5.00 and 6.00, while phototype I was primarily represented in younger female groups (e.g., 17.6% in Lot 1.00). These findings suggest that age and gender may influence the short-term cutaneous response to the applied intervention, while skin phototype distribution remained relatively consistent across study groups.

The intensity of response measured by Test A (20 min post-application) and Test B (24 h) showed group- and age-dependent variability. In female groups, response magnitude tended to increase with age: Lot 1.00 (mean age 20.6 ± 3.5 years) had a Test A mean of 1.06 ± 2.04, while Lot 3.00 (mean age 61.8 ± 10.3 years) exhibited the highest Test A score of 2.13 ± 3.15. A one-way ANOVA confirmed that differences in Test A responses between lots were statistically significant (*p* < 0.01).

A similar pattern was observed for Test B: values increased from 0.29 ± 1.19 in Lot 1.00 to 1.03 ± 2.05 in Lot 3.00, with significant between-group differences (*p* < 0.01, ANOVA).

Among male participants, Lot 5.00 (younger, mean age 24.5 ± 5.8 years) demonstrated the lowest response levels (Test A: 0.88 ± 1.93; Test B: 0.15 ± 0.86), while older males in Lot 6.00 (mean age 54.1 ± 13.8 years) showed increased Test B responses (1.03 ± 2.05), similar to older female participants.

These findings support a significant positive association between age and skin reactivity, as confirmed by Pearson correlation coefficients (r = 0.237 for Test A and r = 0.298 for Test B, both *p* < 0.001) (Table 2).

Notably, no measurable response was recorded at 96 h in any group (Test C = 0.00 ± 0.00), indicating complete resolution of the skin response.

#### 2.3.2. Skin Phototypes According to Gender

The overall distribution of Fitzpatrick skin phototypes among male and female participants is summarized in Table 3. Phototype II was the most prevalent across both genders, observed in 45.6% of males (n = 31) and 52.9% of females (n = 72). Phototype III was more frequently represented in males (38.2%, n = 26) than in females (31.6%, n = 43), whereas phototype I was slightly more common among females (15.4%, n = 21) compared to males (16.2%, n = 11). These findings indicate a relatively similar distribution pattern across genders, with a predominant representation of skin phototype II, followed by phototype III, and a lower prevalence of phototype I. This distribution reflects a typical range expected in populations of predominantly light to medium skin tones and is important for interpreting differential cutaneous responses observed in the time-dependent tests.

Figure 1 illustrates the distribution of Fitzpatrick skin phototypes (Types I–III) among male and female participants. Overall, skin phototype II was the most prevalent in both genders. Among females, phototype II was particularly dominant, with a count approaching 70 individuals, whereas males showed a lower but still significant count of approximately 30 individuals in this category. Phototype III was the second most frequent in both sexes, with around 43 females and 26 males. Phototype I was the least represented, especially among males, with only about 11 individuals, while female participants had a slightly higher count of approximately 21.

These data confirm that most participants had skin phototypes II and III, with phototype II being especially common among females. This distribution is relevant for interpreting skin-related responses to the tested interventions, as Fitzpatrick phototype can influence skin sensitivity and reactivity.

#### 2.3.3. Skin Phototype by Age

The mean age of participants was analyzed according to their Fitzpatrick skin phototype. Individuals with phototype I had a mean age of 36.50 years (±16.50), while those with phototype II had a slightly higher mean age of 39.63 years (±18.34). Participants with phototype III showed a comparable mean age of 38.25 years (±15.99).

Overall, the age distribution was relatively consistent across phototypes, with no significant age-related clustering by skin type. This suggests that the distribution of Fitzpatrick skin phototypes within the study population was not strongly influenced by age, allowing for more balanced comparisons of skin responses across phototypes without age acting as a major confounder (Table 4).

Figure 2 presents the age distribution of participants stratified by Fitzpatrick skin phototype (Types I–III) using a box-and-whisker plot. The median age appears relatively similar across all three phototypes, falling between approximately 30 and 40 years.

The interquartile range (IQR)—representing the middle 50% of ages—also shows comparable spread among the three groups. Phototype II has the widest range, with ages extending from below 20 up to over 85 years, indicating a broader age distribution. Phototypes I and III show slightly narrower ranges but still include participants from late adolescence to older adulthood. There are no evident age outliers, and the data confirm that all three phototype groups include participants from a wide age spectrum. This supports the representativeness of the sample and reduces the likelihood that age alone explains any differences observed in skin responses among phototypes.

#### 2.3.4. Skin Phototype by Groups

The distribution of Fitzpatrick skin phototypes across the six experimental groups (Lots 1.00 to 6.00) is presented in Table 5. Overall, skin phototype II was the most frequently represented type in each group, though the differences between phototypes were relatively modest. Specifically, phototype II ranged from 14.6% in Lot 6.00 to a peak of 20.4% in Lot 3.00. Phototype III followed closely in frequency, showing the highest prevalence in Lots 5.00 and 6.00 (each 18.8%). Phototype I showed a relatively even distribution, ranging from 12.5% in Lot 3.00 to 18.8% in Lots 1.00, 4.00, and 6.00.

Each group showed a balanced representation of the three phototypes, without any group being strongly skewed toward one specific type. This homogeneity across experimental groups supports the assumption that skin phototype was evenly distributed, reducing the potential for bias in skin reaction outcomes attributed to phototype variability. It also allows for more reliable comparisons between groups in subsequent analyses.

Although descriptive differences in skin response scores were observed between experimental lots—particularly higher Test A and B values in older participants—one-way ANOVA did not show statistically significant differences between groups (*p* = 0.217 for Test A, *p* = 0.119 for Test B).

Post hoc Tukey HSD tests further confirmed the absence of significant pairwise differences between any two lots (all *p*-adj > 0.05). This suggests that while age and phototype showed significant associations with skin reactivity (see correlation results), lot assignment alone did not significantly influence test outcomes.

Figure 3 shows the distribution of Fitzpatrick skin phototypes (Types I–III) across the six experimental groups (Lots 1.00–6.00). Skin phototype II was the most prevalent in all groups, with the highest count observed in Lot 3.00 (n = 21). Phototype III was also well represented and relatively evenly distributed among groups. Phototype I had the lowest frequency across all lots, with counts ranging between 4 and 6. Overall, the skin phototype distribution was consistent across experimental groups, supporting group comparability. As shown in Figure 3, the majority of participants in Lot 1 presented with Phototype II.

#### 2.3.5. Relationship

Figure 4 presents a relationship map visualizing the interconnectedness among three key variables: Skin Phototype (Fitzpatrick classification), Test A (20 min post-application), and Test B (24 h post-application). Each node represents a specific category within a variable, and the size of the nodes reflects the category count (number of occurrences in the dataset). Edges (lines) between nodes represent relationships or co-occurrence frequencies, with line thickness indicating the strength or frequency of the relationship (i.e., higher co-occurrence = thicker line).

Notably, the largest nodes correspond to categories with the highest frequency: TestB_24 h = 0 and Skin Phototype = 2, indicating these were the most common values among participants. The strongest connections (thickest lines) were observed between these high-frequency categories and several TestA values, highlighting patterns in skin response over time. For example, TestA values around 0 and 1 were strongly connected to Skin Phototype 2 and TestB = 0, suggesting a correlation between early skin reaction intensity and phototype, with resolution by 24 h.

The map also illustrates weaker connections to less common categories such as TestA = 15 or Phototype = 1, indicated by smaller nodes and thinner edges. Overall, this network-style visualization highlights dominant patterns in the data and suggests that individuals with phototype II most frequently exhibited mild initial responses (TestA) that resolved within 24 h (TestB = 0).

#### 2.3.6. Pearson Correlation

Pearson correlation analysis revealed several statistically significant associations among the measured variables (Table 6). A moderate positive correlation was observed between age and both TestA_20 min (r = 0.237, *p* < 0.001) and TestB_24 h (r = 0.298, *p* < 0.001), indicating that older participants tended to exhibit stronger skin responses at both 20 min and 24 h post-application.

Additionally, TestA_20 min and TestB_24 h were strongly positively correlated (r = 0.346, *p* < 0.001), suggesting that individuals with an early skin reaction were more likely to continue showing a response at 24 h.

Importantly, skin phototype (Fitzpatrick classification) showed significant inverse correlations with both TestA_20 min (r = −0.369, *p* < 0.001) and TestB_24 h (r = −0.317, *p* < 0.001). These findings indicate that individuals with lighter skin phototypes (e.g., Type I) were more likely to develop stronger early and persistent skin responses, whereas darker phototypes were associated with lower reaction intensity.

No significant correlations were found with gender or TestC_96 h, the latter of which had constant values and was thus excluded from the correlation matrix.

Figure 5 Linear regression plots showing the relationship between age and skin response scores:

A weak positive correlation was observed between Test A (20 min) and age (R^2^ = 0.056), with the regression equation y = 36.27 + 1.79·x.

A slightly stronger positive correlation was noted between Test B (24 h) and age (R^2^ = 0.089), with the regression equation y = 36.67 + 3.03·x.

These findings suggest that older participants tended to show higher skin response values, particularly at 24 h.

#### 2.3.7. Group-Level Variation in Response Categories or Skin Phototypes

Figure 6, Bar charts showing the subgroup distributions for each experimental lot (Lots 1 to 6), comparing the values for individual subgroups (blue) against the entire sample (white/whitesmoke) for three categories (e.g., Fitzpatrick skin phototypes or response scores).

Lot 1: A distinct peak is observed in the second category (middle bar), exceeding both the first and third categories, with subgroup values (blue) clearly higher than the total sample average.

Lot 2: The same general pattern is maintained, with the second category highest. The third category is also high, but the first remains low, closely aligned with total sample averages.

Lot 3: The subgroup in the second category exceeds the total sample (white top), while the third also shows moderate elevation. The first category remains lowest, similar to Lot 2.

Lot 4: All three categories show moderate-to-high subgroup representation, with the second again dominating. Slight increases are seen over the total sample average in categories 1 and 3.

Lot 5: The third category shows the largest deviation from the total sample, with a higher bar for the subgroup, followed by the second and then the first.

Lot 6: A similar trend to Lot 5, with consistently high subgroup values across the second and third categories and slight elevation in the first.

These charts visually confirm that the second category (middle bar) was the most consistently represented across all lots, while variation was most noticeable in the first and third categories, particularly in Lots 5 and 6. The use of subgroup coloring (blue) allows comparison against the overall sample distribution (whitesmoke).

#### 2.3.8. Correlation Between Demographics, Phototype, and Skin Response

To further explore the relationships among participant characteristics and skin reactivity, we conducted a Pearson correlation analysis focusing on gender, age, Fitzpatrick skin phototypes, and test responses at 20 min (Test A) and 24 h (Test B). While gender showed no significant associations, age and skin phototype demonstrated meaningful correlations with skin response intensity. Specifically, older participants tended to exhibit stronger reactions, and lighter phototypes were more prone to heightened cutaneous responses. These statistically significant correlations (*p* < 0.01) are visually summarized in the heatmap below, where only the significant relationships are displayed for clarity (Figure 7).

Age showed moderate positive correlations with both TestA_20 min (r = 0.24) and TestB_24 h (r = 0.30), indicating stronger skin reactions in older participants. TestA_20 min and TestB_24 h were strongly correlated (r = 0.35), suggesting consistency of reaction over time. Skin phototype was negatively correlated with both TestA_20 min (r = −0.37) and TestB_24 h (r = –0.32), reflecting stronger reactions in lighter skin types.

These correlations highlight age and phototype as important predictors of early and short-term cutaneous responses.

Overall, the results demonstrated that age and skin phototype significantly influenced short-term cutaneous responses, with older participants and those with lighter phototypes exhibiting higher Test A and B scores. Gender differences were minimal, and no persistent responses were observed beyond 24 h. The consistent distribution of Fitzpatrick skin types across groups supports the internal validity of the comparisons.

## 3. Discussion

According to the specialized literature, *Origanum vulgare* L. oil contains a multitude of volatile compounds from different classes, compounds that have been proven to have multiple pharmaceutical effects [10]. Among the most frequently encountered volatile compounds, we can mention the following: monoterpenes such as carvacrol and thymol. Also, in addition to these representative compounds and to which well-known therapeutic effects are associated (antimicrobial, antitumor, anti-aging, healing activity), there are also acyclic or cyclized compounds such as linalool, myrcene, terpinene, α-pinene, these being highlighted in the oil in smaller quantities [6,10].

In the studies carried out so far, classes of compounds such as flavonoids, tannins and sesquiterpenes have also been identified, being attributed mainly to the antioxidant properties of oregano oil [11].

According to recent studies, it can be stated that the concentration and types of volatile compounds may vary depending on the type of extraction and the oregano plant used, but the medicinal properties of the plant are unanimously confirmed [12,13].

The shortcomings of volatile oils are also well known, namely the fact that they are extremely sensitive to environmental and storage factors.

In addition, applying them to the skin in oily form can be risky, with irritating effects and accumulation in the dermis being mentioned [2]. Thus, to improve bioavailability and increase the stability of oregano oil, inclusion in a vehicle system is mandatory [14].

The emulgel formulation in this study, which included oregano oil, is the perfect solution to solve pharmacokinetic and stability problems.

The percentage of oregano oil included in the preparation formula is 0.5% (0.005 mg/mL), this value being considered non-toxic. Some studies consider values between 0.015 mg/mL–0.09 mg/mL of oregano oil to be safe, the recommended concentration for application to the dermis with a subcytotoxic value being 0.004–0.006 mg/mL of oil [3,9,15,16].

The emulgel formulation was made using compounds that act as absorption promoters (apricot oil) and compounds, such as ceramides and peptides, that will improve anti-aging properties and collagen synthesis in the skin, which will also have a regenerative role for the skin barrier [17].

Being an emulgel, the oily phase and the water phase are emulsified with an emulsifying agent, namely olive wax. Propanediol acts as a humectant and dispersing agent, and after dermal application of the emulgel formed, it acts as a promoter of dermal penetration. To avoid a possible alteration of the emulgem, fragard was used in the preparation formula, which also has antimicrobial properties. The hydrophilic excipient was represented by purified water [18,19].

The antimicrobial properties of oregano oil are mentioned in the literature, highlighting the fact that it acts both as an antibacterial and bactericidal against several bacterial strains such as *Staphylococcus aureus*, *Escherichia coli*, *Pseudomonas aeruginosa*, etc. [9,20].

These bacterial strains are frequently found in the affected dermis in pathologies such as acne, dermal infections, dermal lesions, inflammatory diseases, etc.

*Staphylococcus aureus* is responsible at the dermis level for the release of toxins that will mediate an inflammatory response due to the increase in the level of proinflammatory cytokines. This process can lead to the appearance of boils, folliculitis, ulcerations or wounds in the superficial and or deep skin tissue [21,22].

Among the Gram-negative bacteria, the most sensitive to oregano oil according to recent studies is *Pseudomonas aeruginosa* [23].

Testing the OvO and OvOF emulgels from the point of view of antimicrobial properties demonstrated the existence of their antibacterial and bactericidal effect at different time intervals (Table 1).

The antimicrobial activity exerted by the OvOF emulgel can be attributed to the antimicrobial compound present in the formulation, namely Fragard. However, it can be stated that the antimicrobial and even bactericidal effect is present in the formula containing oregano oil (OvO), these properties being attributed to the oil included in the emulgel.

It can also be stated that for both tested samples and in the case of both bacterial strains, a gradual decrease in the bacterial load is observed, but all these effects are more intense and faster in the case of the OvO formulation.

In the case of the OvOF emulgel, a residual bacterial growth was observed at 12 h, which did not happen in the case of the OvO formula.

Moreover, the OvO emulgel showed a bactericidal action on the Gram-positive bacterial strain at time T3 (2 h), as well as at time T2 (1 h) in the case of the Gram-negative bacteria.

The antibacterial and bactericidal mechanism of oregano oil on Gram-positive bacteria such as *Staphylococcus aureus* is based on the major phenolic compounds (carvacrol and thymol) present in the oil and their hydrophobic characteristics. In practice, they have an increased affinity for the hydrophobic elements in the structure of the bacterial wall, thus increasing its permeability, eventually destroying it and causing the loss of genetic material, destroying it [2,24].

Gram-negative bacteria are characterized by the existence of a uniform, external layer formed by lipopolysaccharides that gives them protection and increases their pathogenicity. Oregano oil has the ability to destroy this coating, which completely destroys the bacteria.

The antibacterial activity observed for OvO emulgel correlates with its carvacrol (≈68%) and thymol (≈12%) content, both being phenolic monoterpenes with well-known membrane-disruptive effects. Incorporating oregano oil into an emulgel matrix enhanced stability against oxidation and evaporation, provided controlled release through the hydrogel network, and significantly reduced direct dermal exposure that may cause irritation—advantages consistent with prior reports on nanoemulsions and polymeric gels containing oregano oil [6,14]. In contrast to simple ointments or nanoemulsions, the emulgel system offers a biphasic carrier that balances lipophilic solubilization and aqueous dispersion, resulting in improved bioavailability and cutaneous tolerance.

Among other bacteria and fungi involved in dermatological diseases and on which oregano oil has an antimicrobial effect mentioned in the literature are as follows: *Staphylococcus epidermis*, *Escherichia coli*, and *Trichophyton rubrum* [2,25,26].

Following the favorable results regarding the antimicrobial capacity of OvO emulgel, the OvO emulgel formula could be an effective alternative for the treatment of dermal bacterial infections (especially since the mentioned bacteria are known to be resistant to conventional treatments).

The formulation was further tested in terms of its possible irritating effect on the skin.

The choice of the test site was made taking into account anatomical, physiological and microbiological aspects.

Thus, we chose to apply and test the OvO emulgel compared to OvOF on the forearm, on its volar surface, to the detriment of the classic testing (posterior thorax region).

This choice is based on a series of anatomical, physiological and microbiological differences between the previously mentioned skin regions, which together significantly influence the absorption of the product, as well as the reproducibility of the results.

Influence of sebum and gland density

The skin on the posterior thorax is a very dense area in sebaceous glands, with an average density of around 100 glands/cm^2^, and the skin of the forearm (volar surface) has a much lower density of a few dozen glands/cm^2^. This leads to a lower sebum production on the forearm, which leads to a reduced interaction of endogenous lipids with topically applied substances. The increased sebum present in seborrheic areas can reduce the effectiveness of antimicrobial agents, either by forming a protective lipid biofilm for bacteria or by chemically fusing with the active substances. Choosing to test on the forearm minimizes these influences, which provides a more optimal environment for evaluating antimicrobial effect [27,28].

Reduced bacterial colonization and stable microbial flora

The posterior chest is an area frequently colonized by lipophilic bacteria, such as *Cutibacterium acnes*, *Staphylococcus aureus* and *Malassezia*, due to the persistent seborrheic environment. *Staphylococcus epidermis* and other commensal Gram-positive cocci (*Micrococci, Corynebacteria*) predominate on the forearm, representing >90% of the resident commensal flora at this level. However, the forearm area is considered to be a “dry” area that presents a lower colonization, and the commensal flora is milder. Taking into account these differences in colonization for testing an antimicrobial product, the forearm offers us an environment with a lower commensal flora, with the microbial “background noise” being reduced.

Biophysical properties that favor controlled absorption

Taking into account the anatomical region, several studies have demonstrated variations in the thickness of the stratum corneum and the parameters of the skin barrier. The stratum corneum of the forearm has an average thickness of approximately 18 µm, demonstrating a relatively uniform skin barrier, compared to the posterior thorax region where the average thickness of the stratum corneum is approximately 11 µm. Hydration of the stratum corneum and transepidermal water loss differ depending on the topographic region: the forearm area is considered a “dry” area (poor in glands), and tends to have a lower cutaneous water loss, while areas that are rich in sebaceous glands (forehead or upper chest) show a much higher cutaneous water loss. This suggests a lower skin permeability in the forearm compared to areas with richer sebaceous glands, due to the much better-organized corneum barrier [24,28,29].

Increased accessibility and reproducibility

The forearm area is much more easily accessible, both for applying the emulgel and for repeated evaluations, namely photographs, sample collection and instrumental measurements. The location of the forearm allows for simultaneous application of the OvO product and an OvOF control, which can reduce interindividual variability.

By testing the innovative emulgel product with oregano oil included on the skin, we wanted to highlight any possible irritation reactions, as these are undesirable. The lack of irritating reactions represents a success in terms of the emulgel formulation with oregano oil and confirms the safe application of the created product on the skin.

In this study, we investigated the influence of demographic characteristics and Fitzpatrick skin phototype on short-term cutaneous responses across six experimental groups. Our findings demonstrate that while gender did not significantly affect response outcomes, both age and phototype played a decisive role.

Older participants exhibited stronger reactions at 20 min and 24 h, and lighter skin phototypes were consistently associated with more intense responses. Conversely, no measurable reactions were observed at 96 h, indicating complete resolution of the skin changes by this time point.

Although descriptive trends suggested variability between experimental lots, statistical analysis confirmed that lot assignment itself was not a significant factor. Instead, individual characteristics—particularly age and skin phototype—were the main determinants of reaction intensity. These results highlight the importance of considering participant demographics in dermatological and cosmetic safety testing, as they can meaningfully shape early skin responses. Future research should validate these findings in larger and more diverse populations, while also exploring potential biological mechanisms underlying the observed phototype- and age-dependent differences.

## 4. Materials and Methods

The reagents used in this study were purchased from Elemental (Oradea, Romania), Farmachim 10 SRL (Ploiesti, Romania), Sigma-Aldrich (Taufkirchen, Germany), and Fluka (Buchs, Switzerland). The compounds used are of adequate purity, attested by analysis bulletins issued by the manufacturer. *Origanum vulgare* L. oil (no. 60204665) is purchased from doTERRA, accompanied by chemical characterization.

### 4.1. Formulation and Organoleptic Control of Emulgel with Oregano Oil (OvO)

The *Origanum vulgare* L. essential oil (lot no. 60204665) was purchased from doTERRA (USA). According to the supplier certificate, the oil was obtained by steam distillation from flowering aerial parts of *O. vulgare* L. (chemotype rich in carvacrol and thymol, harvested in Albania, 2024). Gas-chromatography–mass-spectrometry (GC-MS) analysis confirmed carvacrol (25.59%) and thymol (3.11%) as major constituents. The complete table of compounds is provided in Appendix A.

The percentages of the major components were calculated using Formula (1):(1)$$Compound\left(\%\right)=\frac{peak\;area\;of\;compound}{total\;peak\;areas}\;\times 100$$

The steps for formulating the innovative emulgel preparation with oregano oil included are highlighted in Figure 8.

The emulgel formulation was made according to Table 7 by preparing two emulgel formulas: one without oregano oil included (OvOF) and one with oregano oil included (OvO).

The aqueous phase of the emulgel was composed of xanthan gum (2%) and propanediol (3%), which were moistened with distilled water.

The oily phase of the emulgel was formulated from melted olive wax (5%) in which apricot oil (7%) was incorporated, as well as ceramides (2%) and peptides (1%). After homogenization, Fragard (1%) was added, and 30 drops of Oregano oil were added to the OvO formula (equivalent to the final concentration of the emulgel of 0.5%).

The emulsion phases were homogenized using a rotor–stator homogenizer (IKA T25 Digital Ultra-Turrax, Staufen im Breisgau, Germany) operating at 10,000 rpm for 10 min at 25 ± 1 °C. Olive wax was melted at 70 °C before incorporation of the oily phase. The aqueous phase was slowly added to the oil phase under continuous stirring (500 rpm) to obtain a uniform emulsion.

After formulation, organoleptic control was performed (evaluating odor, color, consistency) and the pH of the emulgel was measured using a Hanna Instruments Inc.8417 pH meter, Woonsocket, RI, USA, whose probe was immersed directly in the sample after prior calibration.

The stability of the emulgel was determined by centrifugation process using the centrifuge Hettich Universal 320 R (Tuttlingen, Germany). A total of 20 mL of the formula was centrifuged in conic polyethylene tubes at 6000 RPM for 20 min at laboratory temperature (25 ± 1) °C. Time-dependent phase separation was simulated by the centrifugation process, and the phase stability of the emulgels was evaluated by the stability index (*SI*). *SI* was calculated by Equation (2):*SI* (%) = (1 − *V*_2_/*V*_1_) × 100(2)
where *V*_2_ is total volume of the emulgel and *V*_1_ is the volume of oil phase (supernatant) after the centrifugation [30].

### 4.2. Determination of the Antimicrobial Capacity of OvO Emulgel

The antibacterial activity of OvO and OvOF emulgels was evaluated against *Staphylococcus aureus* NCTC 12493 and *Pseudomonas aeruginosa* ATCC 27853 using a modified minimum inhibitory concentration (MIC) and minimum bactericidal concentration (MBC) assay, adapted from CLSI M07-A11 guidelines. Briefly, bacterial suspensions were adjusted to 0.5 McFarland (≈1.5 × 10^8^ CFU/mL) in sterile saline. Each emulgel sample (0.9 mL) was mixed with 0.1 mL of standardized inoculum (final ratio 9:1, corresponding to 90% *v*/*v* emulgel) to obtain a total test volume of 1 mL. Control tubes contained 0.1 mL inoculum plus 0.9 mL saline. After vortexing to ensure homogeneity, mixtures were incubated at 37 °C. At defined time points (0, 30 min, 1, 2, and 24 h), 100 µL aliquots were serially diluted and plated in triplicate on Columbia blood agar. Plates were incubated 24 h/37 °C, and viable counts expressed as CFU/mL (±SD). The MBC was defined as the lowest concentration causing ≥99.9% reduction in CFU compared with the initial inoculum. Gentamicin (10 µg/mL) served as positive control.

All tests were performed in triplicate.

### 4.3. In Vivo Testing of the Irritability of OvO Emulgel

Testing of possible irritant reactions of OvO emulgel was performed in an authorized dermatology office with the consent of the patients included in the study. The testing method consisted of applying occlusive patches containing 100 µL of OvO emulgel (compared to OvOF emulgel) and which was subsequently applied in a horizontal position on the forearm.

The testing was performed only on patients whose test area did not present eschar, lesions or other skin conditions (Figure 9).

The testing was carried out on 6 experimental groups that were divided according to sex, age and skin type. The skin response (degree of irritability, redness and inflammation) to emulgel was monitored at three different times: Test A = 20 min, Test B = 24 h, Test C = 96 h.

Occlusive patches consisted of Finn Chamber^®^ aluminum disks (diameter 8 mm, area 0.5 cm^2^) mounted on hypoallergenic adhesive tape. Each chamber received 100 µL of emulgel measured with a calibrated micropipette to standardize dose density (200 µL/cm^2^). Patches were applied to the volar forearm under controlled ambient conditions (22 ± 1 °C, 45 ± 5% RH) for 24 h. A parallel control patch containing base emulgel (OvOF) and one blank patch (no sample) were applied to account for mechanical irritation.

### 4.4. Statistical Analysis

The statistical analysis was accomplished using the software SPSS Statistics v30 (IBM Corp., Armonk, NY, USA). The Bravais–Pearson correlation coefficient determined an independent indicator of the units of measurement for three variables. The statistical analysis interpretation of the results of the in vivo tests was performed using the ANOVA test.

Statistical analysis results are presented in a structured order: (1) group and demographic description, (2) comparative analysis of Test A (20 min), Test B (24 h), and Test C (96 h) between lots, and (3) correlation between demographic variables and response intensity. One-way ANOVA followed by Tukey HSD post hoc tests were used for between-group comparisons, and Pearson correlation coefficients quantified linear relationships. Significant results are summarized in Table 6 and illustrated in Figure 6, Figure 7 and Figure 8.

### 4.5. Ethical Approval

The in vivo study was approved by the Ethics Committee of the University of Oradea (approval no. CEFMF/5, 28 March 2024), in accordance with the Declaration of Helsinki (2013 revision). All participants provided written informed consent before inclusion. Each experimental lot included 34 participants (n = 204 total). Inclusion criteria: healthy volunteers aged 18–70 years, no active dermatoses; exclusion criteria: chronic skin disease, recent topical/systemic antibiotic or corticosteroid use.

## 5. Conclusions

According to the tests carried out, it can be concluded that the OvO type formula is successfully produced, being characterized by a pH (pH = 6) corresponding to application on the skin, as well as increased stability (having a stability index SI = 95%). Microbiological tests have demonstrated antibacterial and even demonstrated antibacterial activity on Gram-positive bacteria (*Staphylococcus aureus*) and Gram-negative bacteria (*Pseudomonas aeruginosa*). Thus, the OvO formula was shown to have a bactericidal effect after 2 h of application for Gram-positive bacteria and for Gram-negative bacteria after one hour of application, the latter being more sensitive. Furthermore, the total eradication of the bacteria on which the testing was carried out was highlighted, with no residual growth after 12 h.

Following in vivo testing of the possible skin irritation and redness reaction, it was demonstrated that regardless of sex, age or skin phenotype, both in the short term (20 min and after 96 h of application), the formulated OvO emulgel did not produce irritating reactions on the skin.

Taking into account these results, it can be concluded that the innovative OvO formula is suitable for use as an alternative treatment in the therapeutic management of bacterial dermal diseases.

As future perspectives, it is desired to test the emulgel from the point of view of the healing and reparative properties of wounds infected with bacteria sensitive to the phytocompounds present in *Origanum vulgare* L. oil.

## Figures and Tables

**Figure 1 pharmaceuticals-18-01768-f001:**
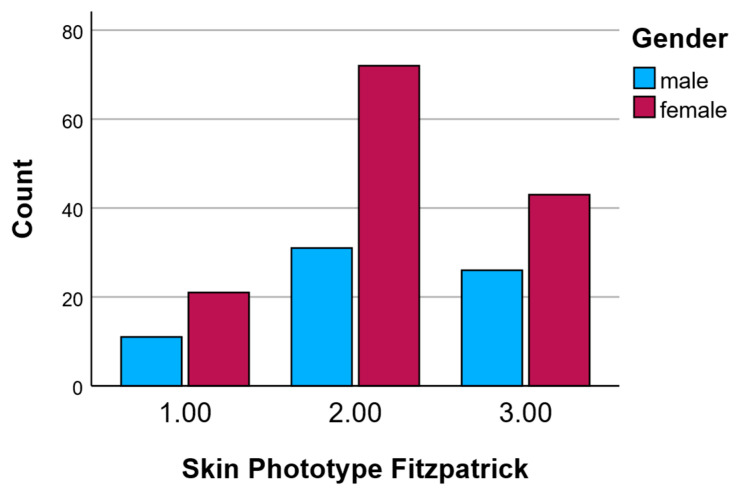
Skin phototypes according to gender.

**Figure 2 pharmaceuticals-18-01768-f002:**
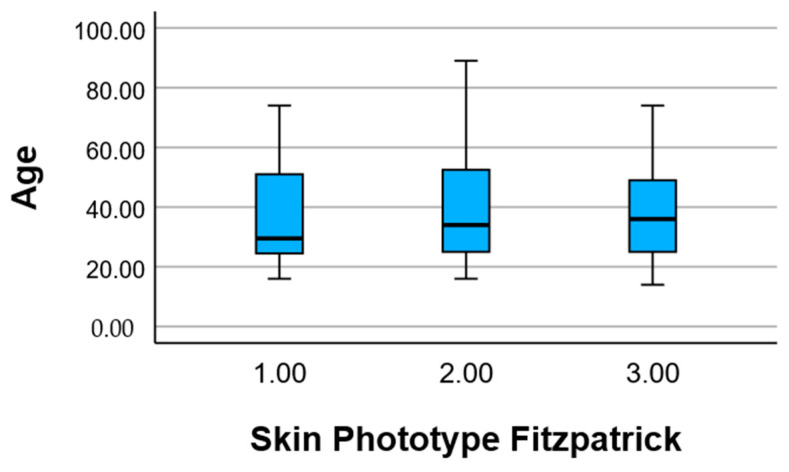
Skin phototypes according to age.

**Figure 3 pharmaceuticals-18-01768-f003:**
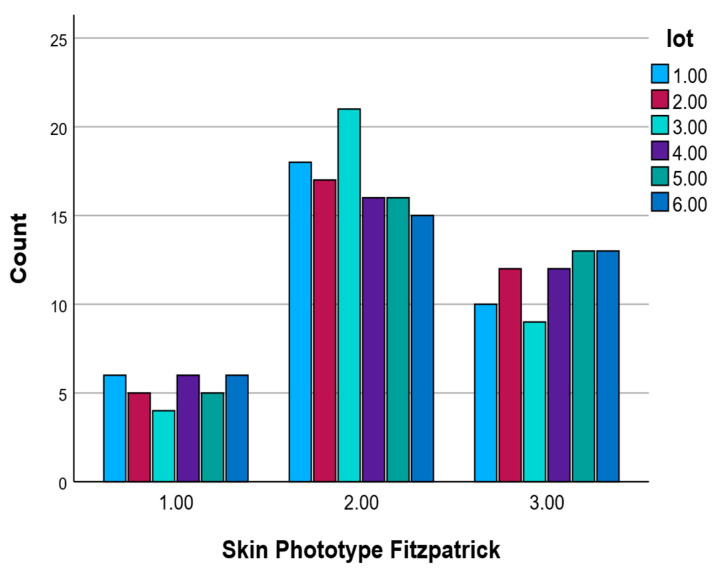
Skin phototype frequencies by experimental group based on the Fitzpatrick classification.

**Figure 4 pharmaceuticals-18-01768-f004:**
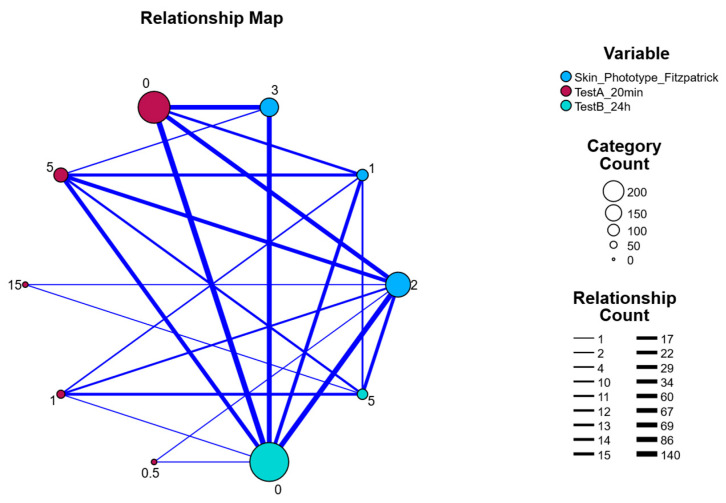
Relationship Map Between Skin Phototype (Fitzpatrick), Test A (20 min), and Test B (24 h) categories with relationship frequency indicated by Edge Thickness.

**Figure 5 pharmaceuticals-18-01768-f005:**
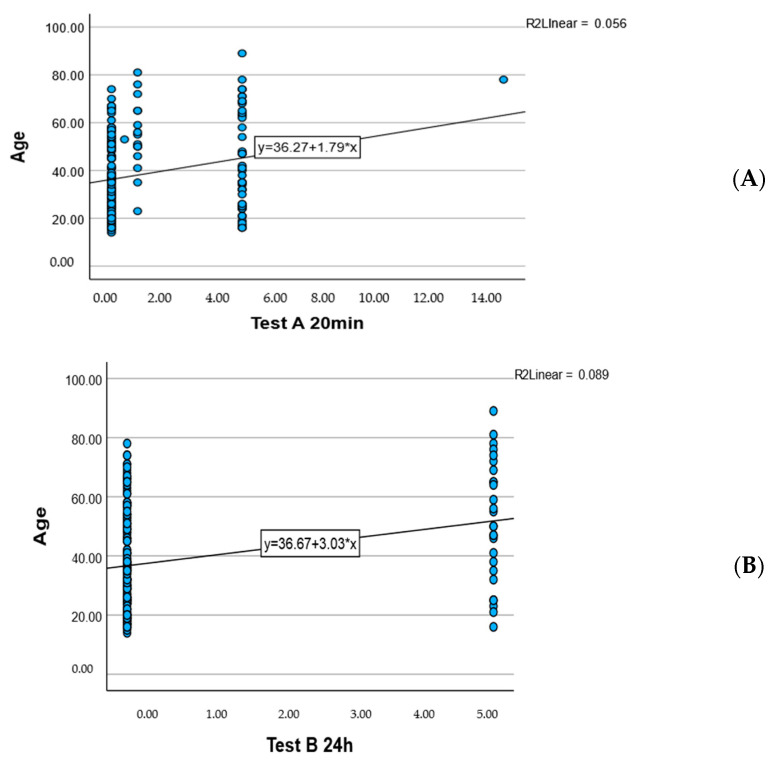
Association Between Participant Age and Cutaneous Reaction Scores at Early (**A**) and Delayed (**B**) Time Points.

**Figure 6 pharmaceuticals-18-01768-f006:**
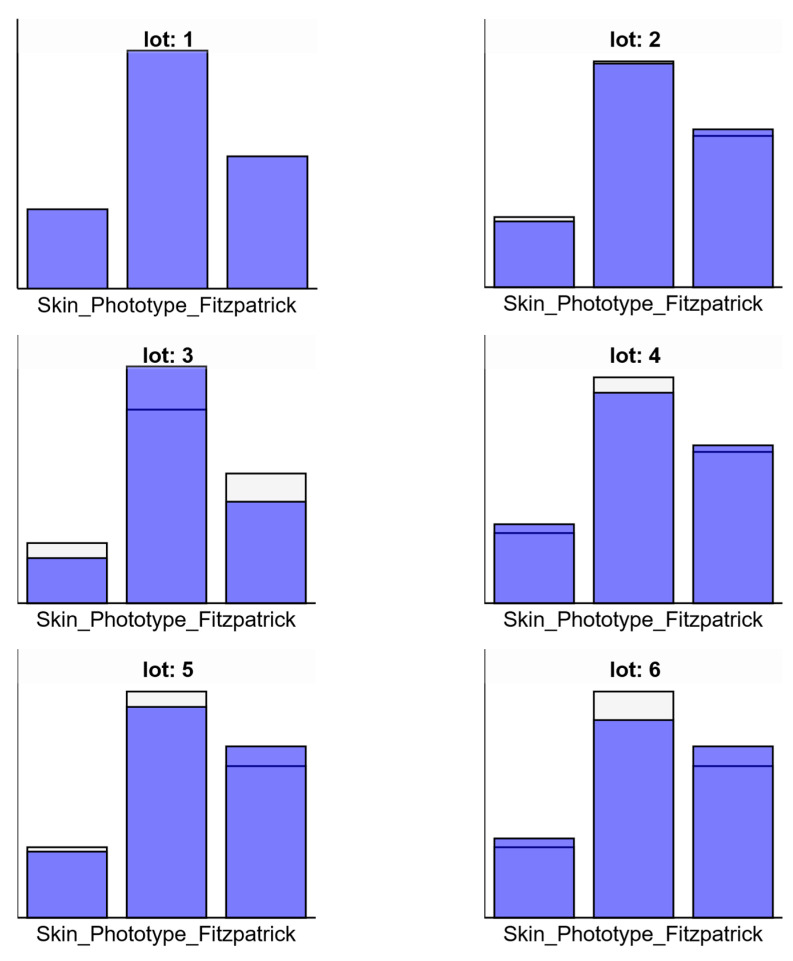
Comparison of subgroup distributions by experimental lot (Lots 1–6) across three categories.

**Figure 7 pharmaceuticals-18-01768-f007:**
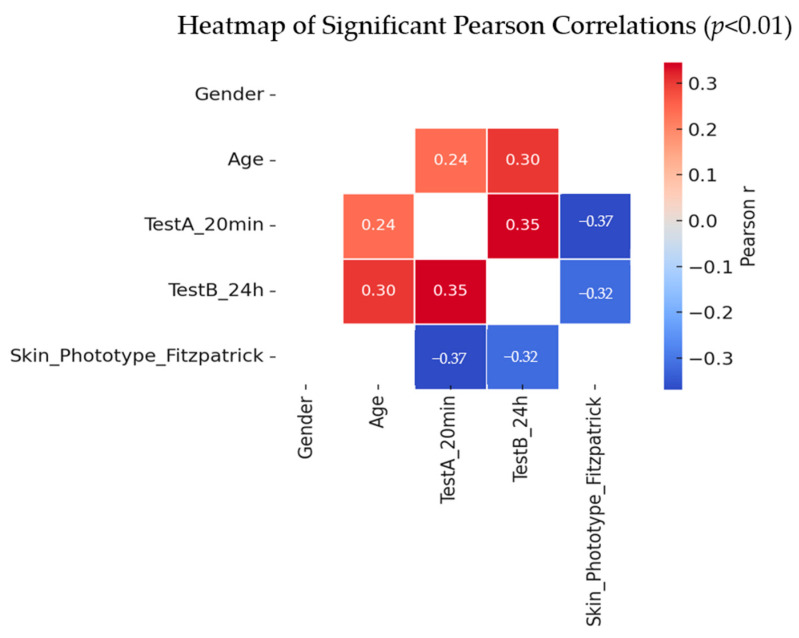
Heatmap showing significant Pearson correlations (*p* < 0.01) between demographic characteristics, skin phototype, and skin response variables.

**Figure 8 pharmaceuticals-18-01768-f008:**
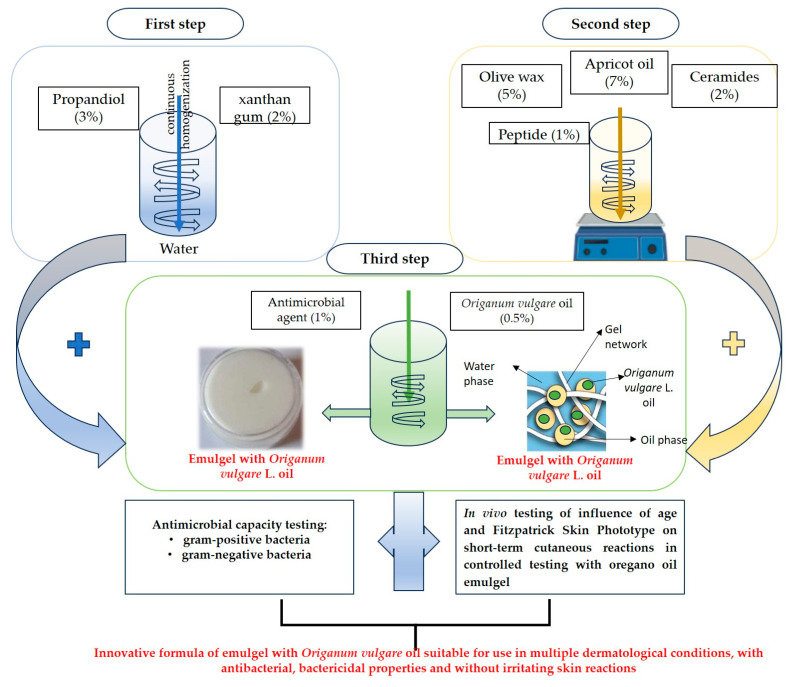
Experimental design of the emulgel formulation steps and its testing in terms of antimicrobial capacity and in vivo testing.

**Figure 9 pharmaceuticals-18-01768-f009:**
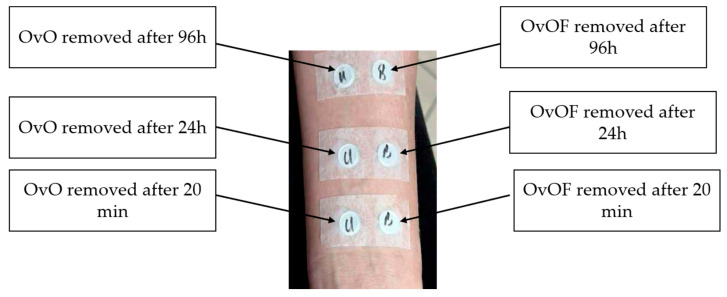
Experimental design for application and removal of patches used for in vivo testing of possible irritant reactions of OvOF and OvO emulgel.

**Table 1 pharmaceuticals-18-01768-t001:** Results of testing the antimicrobial capacity of OvOF and OvO emulgel formulas.

Bacterial Strains Used for Testing	OvOF					OvO				
	Time									
	T0	T1	T2	T3	T4	T0	T1	T2	T3	T4
	UFC/µL									
*S. aureus* NCTC 12493	TNTC	<500	<100	<1	<10	TNTC	TNTC	<100	0	0
*P. aeruginosa* ATCC 27853	TNTC	<200	<100	0	0	TNTC	<10	0	0	0

Values represent mean ± SD from triplicate plates. TNTC = Too Numerous To Count, determined by visual inspection according to CLSI M07-A11 (2023). When individual colonies were discernible, counts were expressed as CFU/mL. T0—time immediately after application to the culture medium of the bacterial suspension mixed with the tested sample, T1—after 30 min, T2—after 1 h, T3—after 2 h, T4—after 24 h from seeding on the culture medium.

**Table 2 pharmaceuticals-18-01768-t002:** Demographic characteristics and Skin Test Responses at 20 min, 24 h, and 96 h in six Experimental Groups.

Parameters			Gender		Age (Mean ± SD)	Test A 20 min (Mean ± SD)	Test B 24 h (Mean ± SD)	Test C 96 h (Mean ± SD)	Skin_Phototype_Fitzpatrick		
			Male	Female					1.00	2.00	3.00
Group	1.00	N	0	34	20.59 ± 3.49	1.06 ± 2.04	0.29 ± 1.19	0.00 ± 0.00	6	18	10
		%	0.0%	100.0%					17.6%	52.9%	29.4%
	2.00	N	0	34	36.18 ± 7.56	1.12 ± 2.03	0.59 ± 1.64	0.00 ± 0.00	5	17	12
		%	0.0%	100.0%					14.7%	50.0%	35.3%
	3.00	N	0	34	61.79 ± 10.25	2.13 ± 3.15	1.03 ± 2.05	0.00 ± 0.00	4	21	9
		%	0.0%	100.0%					11.8%	61.8%	26.5%
	4.00	N	0	34	34.94 ± 8.53	1.50 ± 2.30	0.88 ± 1.93	0.00 ± 0.00	6	16	12
		%	0.0%	100.0%					17.6%	47.1%	35.3%
	5.00	N	34	0	24.47 ± 5.80	0.88 ± 1.93	0.15 ± 0.86	0.00 ± 0.00	5	16	13
		%	100.0%	0.0%					14.7%	47.1%	38.2%
	6.00	N	34	0	54.06 ± 13.75	1.00 ± 1.94	1.03 ± 2.05	0.00 ± 0.00	6	15	13
		%	100.0%	0.0%					17.6%	44.1%	38.2%

**Table 3 pharmaceuticals-18-01768-t003:** Gender-based distribution of Fitzpatrick Skin Phototypes among study participants.

		Gender			
		Male		Female	
		Count	Column N %	Count	Column N %
Skin_Phototype_Fitzpatrick	1.00	11	16.2	21	15.4
	2.00	31	45.6	72	52.9
	3.00	26	38.2	43	31.6

**Table 4 pharmaceuticals-18-01768-t004:** Mean Age and Standard Deviation by Fitzpatrick Skin Phototype.

		Age	
		Mean	Standard Deviation
Skin_Phototype_Fitzpatrick	1.00	36.50	16.50
	2.00	39.63	18.34
	3.00	38.25	15.99

**Table 5 pharmaceuticals-18-01768-t005:** Distribution of Fitzpatrick Skin Phototypes across experimental groups.

		Skin_Phototype_Fitzpatrick					
		1.00		2.00		3.00	
		Count	Column N %	Count	Column N %	Count	Column N %
Groups	1.00	6	18.8	18	17.5	10	14.5
	2.00	5	15.6	17	16.5	12	17.4
	3.00	4	12.5	21	20.4	9	13.0
	4.00	6	18.8	16	15.5	12	17.4
	5.00	5	15.6	16	15.5	13	18.8
	6.00	6	18.8	15	14.6	13	18.8

**Table 6 pharmaceuticals-18-01768-t006:** Pearson Correlation Coefficients between Demographic Variables, Skin Phototype, and Skin Response Measures.

Correlations		Gender	Age	TestA_20 min	TestB_24 h	TestC_96 h	Skin_Phototype_Fitzpatrick
Gender	Pearson Correlation	1	−0.024	0.106	0.031	.^a^	−0.041
	Sig. (2-tailed)		0.729	0.134	0.663	.	0.562
	N	204	204	203	204	204	204
Age	Pearson Correlation	−0.024	1	0.237 **	0.298 **	.^a^	0.017
	Sig. (2-tailed)	0.729		<0.001	<0.001	.	0.811
	N	204	204	203	204	204	204
TestA_20 min	Pearson Correlation	0.106	0.237 **	1	0.346 **	.^a^	−0.369 **
	Sig. (2-tailed)	0.134	<0.001		<0.001	.	<0.001
	N	203	203	203	203	203	203
TestB_24 h	Pearson Correlation	0.031	0.298 **	0.346^**^	1	.^a^	−0.317 **
	Sig. (2-tailed)	0.663	<0.001	<0.001		.	<0.001
	N	204	204	203	204	204	204
TestC_96 h	Pearson Correlation	.^a^	.^a^	.^a^	.^a^	.^a^	.^a^
	Sig. (2-tailed)	.	.	.	.		.
	N	204	204	203	204	204	204
Skin_Phototype_Fitzpatrick	Pearson Correlation	−0.041	0.017	−0.369 **	−0.317 **	.^a^	1
	Sig. (2-tailed)	0.562	0.811	<0.001	<0.001	.	
	N	204	204	203	204	204	204

**. Correlation is significant at the 0.01 level (2-tailed). .^a^ Cannot be computed because at least one of the variables is constant.

**Table 7 pharmaceuticals-18-01768-t007:** Excipients used for the formulation of emulgel with oregano oil included (OvO) and emulgel without oregano oil included (OvOF).

Ingredients	OvOF	OvO
	(%)	
*Origanum vulgare* oil	-	0.50
Peptide	1.00	1.00
Ceramides	2.00	2.00
Apricot oil	7.00	7.00
Xanthan gum	2.00	2.00
Olive wax	5.00	5.00
Fragard	1.00	1.00
Propanediol	3.00	3.00
Distilled water	79.00	78.50

## Data Availability

The original contributions presented in this study are included in the article material. Further inquiries can be directed to the corresponding authors.

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
