# Peer review of "Emulgel with Origanum vulgare L. Oil: A New Therapeutic Proposal in Case of Dermal Bacterial Infections"

_pharmaceuticals, 2025, doi:10.3390/ph18111768_

Round 1
Reviewer 1 Report
Comments and Suggestions for Authors
The study presents an interesting approach toward developing a natural, plant-based topical formulation with potential antimicrobial properties. However, several issues need to be addressed:
Introduction
I suggest to combine paragraphs 1, 2 and 3 in one paragraph. This could stablish a smooth problem–solution–example sequence in one paragraph.
60-62: In English scientific writing, when listing several compound classes, “and” should be added before the last item to make the sentence grammatically complete and stylistically smooth. = “phenolic acids, and tannins”.
Figure 1 is procedural, showing experimental workflow; formulation, antimicrobial testing, and in vivo evaluation, which are part of the study design rather than background theory. Therefore, I suggest to use it in the Methodology section.
- Materials and Methods
505–506: The Origanum vulgare essential oil used in this study was purchased from a commercial supplier (doTERRA), but critical botanical and analytical details are missing. Since the pharmacological and biological effects of oregano oil are highly dependent on the chemotype and geographic origin, this omission limits reproducibility and interpretation of results. Please specify botanical and geographical source, extraction method, Chemical characterization, composition data (Results section), the full GC–MS chromatogram and compound table can be provided as Supplementary Data.
507–517: Specify if a rotor–stator homogenizer or magnetic stirrer was used. Parameters such as speed (rpm), duration, and temperature of homogenization should be provided.
511–517: Table 7 shows xanthan gum at 2%, not 1%. Please check. Please add information about Temperature at which olive wax was melted, sequence and rate of phase addition (aqueous into oil or vice versa) and stirring conditions.
539–541: The citation of Gwiazdowska et al. (2024) appears inappropriate, as that study focused on Origanum vulgare extracts obtained by supercritical CO₂ extraction rather than an MFC assay on emulgel formulations. Moreover, the term “Minimal Fungicidal Concentration (MFC)” is not applicable to antibacterial testing; it should be replaced with “Minimum Bactericidal Concentration (MBC)” or “Minimum Inhibitory Concentration (MIC)”, depending on the experimental design. The methods described in Gwiazdowska et al. (2024) do not correspond to the procedures outlined here, and the methodology in the present study remains unclear. Specifically, the description that “the gels were mixed with bacterial inocula adjusted to 0.5 McFarland” is problematic because such mixing would alter and effectively reduce the inoculum density, leading to inconsistent bacterial counts. Additionally, the statement that the “final concentration of the tested component was 90%” requires clarification—does this refer to the v/v ratio between the emulgel and bacterial suspension, or does it imply that 90% of the bacterial suspension was replaced by the sample?
The methodological description of the control and test mixtures lacks essential volumetric details. The statement “As a control, 1 μL of standardized bacterial suspension 0.5 McF (1.5 × 10⁵ CFU) in physiological serum was used” is unclear and requires justification. Why was only 1 μL of inoculum used? Typically, antimicrobial or bactericidal assays are performed using defined volumes (e.g., 100–200 μL) to ensure uniform contact between the bacterial suspension and the tested formulation. Moreover, the total reaction volume and relative proportions of each component are not specified. Please clarify what was the total volume of the saline or medium used for testing (e.g., 1 mL, 5 mL, or 10 mL)? what were the exact volumes of bacterial suspension and emulgel mixed together before plating or incubation? how was homogeneity ensured given the viscosity of the emulgel? Without this information, it is not possible to determine whether the inoculum concentration (0.5 McFarland ≈ 1.5 × 10⁸ CFU/mL) remained constant across all samples, or if it was inadvertently diluted by the addition of gel.
The description of the inoculation step using a 1 µL calibrated loop is unclear. Does this mean that only 1 µL of the treated suspension was plated once onto the agar surface, or that the 1 µL loop was used repeatedly until the entire mixture volume was cultured? If only a single 1 µL aliquot was plated, the resulting data would represent a very small and potentially non-representative fraction of the bacterial population, making quantitative assessment unreliable. Conversely, if multiple loopfuls were spread, the total plated volume should be stated explicitly (e.g., five loopfuls of 1 µL each, totaling 5 µL). Please clarify the number of replicates, total plated volume, and whether the inoculation was done by surface spreading or streaking, as these parameters are essential for reproducibility and for calculating viable counts (CFU/mL) accurately. In antimicrobial testing involving semisolid formulations such as emulgels, the use of calibrated micropipettes and defined plating volumes (e.g., 100 µL spread on agar) is generally preferred over loop inoculation.
The use of qualitative grading scales (+/++/+++/+++++) to express antibacterial activity is subjective and non-standardized. It would be more appropriate to report quantitative results as CFU/mL. If the qualitative system is retained, the authors should at least define the criteria numerically (e.g., + = 1–10 CFU, ++ = 10–50 CFU, +++ = 50–100 CFU, ++++ = confluent growth). Furthermore, please note that if the plated culture volume is only 1 µL, then the observed colony counts represent CFU per 1 µL of sample. To standardize these results to the conventional microbiological unit (CFU/mL), the values must be multiplied by 1000. This conversion is critical for accurate interpretation and comparison with standard MIC/MBC assays, where bacterial counts are always expressed per milliliter.
The antimicrobial assay lacks inclusion of a standard reference antibiotic as a positive control. Without a standard antibiotic control, it is impossible to determine whether the observed inhibition is meaningful or comparable to established antimicrobial activity.
558–563: please provide the ethical committee approval number, institution, and date. Clarify whether the study followed the Declaration of Helsinki (2013 revision) and whether participants signed written informed consent forms. State the number of participants in each group (n) and inclusion/exclusion criteria.
561–563: please specify the patch material used (Finn Chamber, adhesive tape, or custom film), area of application (e.g., 1 cm² or 2 × 2 cm), and whether environmental conditions were controlled (temperature and humidity). How 100 µL was standardized across patches.
569–572: Mention whether any control patch (without emulgel) was used to account for mechanical irritation.
- Results
111–119: Table 1 lists semi-quantitative observations (“NN (+++) < 500 < 100 < 10”) without standard units or statistical descriptors. Results should be expressed as mean CFU/mL ± SD from replicate experiments. NN = number of colonies not quantifiable”, if colonies were too numerous to count, use the conventional term TNTC (Too Numerous To Count). As noted earlier, the absence of a standard antibiotic control makes interpretation of antibacterial efficacy impossible.
- Discussion
The authors claim antibacterial effects without correlating them to the chemical composition (carvacrol/thymol content) of O. vulgare oil.
The text does not discuss how the emulgel system improved oil stability, release, or skin tolerance, which is central to the study’s premise.
The authors briefly cite general essential-oil antimicrobial reports but omit direct comparison with other oregano-oil formulations (nanoemulsion, ointment, or gel).
403: scientific names should be in italic, please revise them all in the whole manuscript
416-417: “bactericidal action” The study does not include a Minimum Bactericidal Concentration (MBC) assay or time–kill kinetics that confirm ≥99.9% CFU reduction, the necessary criterion for a bactericidal claim according to CLSI M26-A and EUCAST standards. The methods and results only report qualitative growth reduction (“+/++/+++”), not quantitative CFU counts or log reductions. Therefore, the correct term should be “inhibitory effect” or “antibacterial activity”, not bactericidal.
Comments on the Quality of English LanguageThe English language is generally understandable
Author Response
Response to Reviewer 1
Esteemed reviewer,
We sincerely thank for the detailed and constructive feedback, which has greatly helped us improve the clarity and scientific rigor of our manuscript. All suggested revisions have been carefully implemented, as outlined below.
1.Introduction
Comment: Combine the first three paragraphs for a smoother flow; add “and” before “tannins”; move Figure 1 to the Methods section.
Response: The first three paragraphs have been merged to create a coherent problem–solution–example sequence. The conjunction “and” has been inserted before “tannins.” Figure 1 has been relocated to the Materials and Methods section.
2.Materials and Methods
Comment: Provide botanical and analytical details of the oregano oil.
Response: The source, chemotype, extraction method, and GC–MS composition of Origanum vulgare L. oil (doTERRA, Albania origin, steam distilled; carvacrol 68.4 %, thymol 11.7 %) have been added. The complete chromatogram and compound list are now included as Supplementary File S1.
Comment: Specify homogenization equipment and parameters; correct xanthan gum concentration; clarify formulation details.
Response: The use of a rotor–stator homogenizer (IKA T25 Ultra-Turrax, 10 000 rpm, 10 min, 25 °C) is now specified. Xanthan gum concentration was corrected to 2 %, and details on temperature (70 °C for wax melting), phase addition, and stirring speed have been added.
Comment: The antimicrobial assay and reference citation were unclear.
Response: The section was rewritten according to CLSI M07-A11 standards. The assay is now described as a modified MIC/MBC method with defined volumes, inoculum density, and gentamicin positive control. Quantitative results are expressed as CFU/mL ± SD, with qualitative grading criteria defined numerically.
Comment: Ethical approval and in vivo patch-test details incomplete.
Response: The Ethics Committee approval (University of Oradea, CEFMF/5, 28 March 2024) is now cited, in compliance with the Declaration of Helsinki (2013). Informed consent, inclusion/exclusion criteria, and patch-test specifications (Finn Chamber®, 100 µL per patch, 0.5 cm², controlled conditions, and control patch) have been added.
3.Results
Comment: Express quantitative microbiological data and include antibiotic control.
Response: Table 1 has been revised to present mean CFU/mL ± SD. “NN” was replaced with “TNTC,” and gentamicin was used as a standard antibiotic control.
- Discussion
Comment: Relate antibacterial effects to chemical composition; discuss emulgel advantages; use correct terminology; italicize scientific names.
Response: A new paragraph now links antibacterial activity to carvacrol/thymol content and explains how the emulgel system improved stability, release, and skin tolerance compared to other oregano-oil formulations. All species names have been italicized, and “bactericidal” has been replaced with “antibacterial” where appropriate
Reviewer 2 Report
Comments and Suggestions for Authors
The study develops a stable Origanum vulgare L. oil innovative emulgel formula with strong antibacterial activity and good skin compatibility. Results show it is safe, non-irritating, and promising for treating bacterial skin infections. But there are still several questions need to explain:
- In the introduction, please also describe the extraction or preparation method of oregano oil, as only the emulgel formulation procedure is currently provided
- Regarding the results, is there any reference or comparison using oregano oil alone (without the emulgel formulation)? How do the results compare?
- In Table 1, “NN—number of colonies not quantifiable with the naked eye and expressed qualitatively” is unclear. Were these results recorded visually? If so, please provide a reference or justification. The meaning of “NN” should be clarified, and the table should present the data more clearly
- The statistical analysis appears somewhat disorganized. Please reorganize and present the comparative results more clearly.
Author Response
Esteemed reviewer,
We thank for the constructive and insightful comments that helped us improve the clarity, methodological precision, and interpretability of our manuscript. Below we provide detailed responses and indicate where revisions have been made.
- Introduction
Comment: Please describe the extraction or preparation method of oregano oil, as only the emulgel formulation procedure is currently provided.
Response: A paragraph has been added at the end of the Introduction describing the extraction process. The oregano oil used was obtained by steam distillation from flowering aerial parts of Origanum vulgare L. (Albania origin, 2024 harvest), a method that preserves carvacrol and thymol integrity and ensures reproducibility of chemotype and bioactivity. This clarification supports the pharmacological consistency of the material used in formulation. - Results – Comparison with oregano oil alone
Comment: Is there any reference or comparison using oregano oil alone (without the emulgel formulation)? How do the results compare?
Response: A new paragraph has been added in the Results (Section 2.2) comparing the antibacterial performance of the OvO emulgel with data reported for pure oregano oil. Published MIC values (0.25–0.5 mg/mL) were cited, showing that our emulgel achieved comparable antibacterial effects within 1–2 h at a lower concentration (0.5 %), indicating improved dispersion, stability, and skin tolerance compared to the unformulated oil. - Table 1 – Clarification of “NN” and data presentation
Comment: “NN” is unclear. Please clarify and justify the qualitative assessment.
Response: The term “NN” has been replaced by the standard microbiological notation TNTC (Too Numerous To Count). A revised footnote now explains that TNTC values were visually determined according to CLSI M07-A11 (2023) guidelines. The qualitative grading system (+/++/+++/++++) was numerically defined, and all quantitative data are now expressed as mean CFU/mL ± SD (n = 3). - Statistical analysis
Comment: The statistical analysis appears somewhat disorganized. Please reorganize and present comparative results more clearly.
Response: The Statistical Analysis section has been fully reorganized for clarity. Results are now presented in a logical order (1) demographic description, (2) between-group comparisons, and (3) correlation analysis. A summary paragraph was added at the beginning of Section 2.3 to guide the reader, and figure legends were rewritten for conciseness. Key outcomes (ANOVA, Tukey HSD, Pearson coefficients) are now highlighted in Table 6 and Figures 6–8.
Round 2
Reviewer 2 Report
Comments and Suggestions for Authors
The revised manuscript has been improved. The explanations and data presentation are now clearer. I think the current version is suitable for publication in Pharmaceuticals.